# Which Commodity Sectors Effectively Hedge Emerging Eastern European Stock Markets? Evidence from MGARCH Models

**Amel Melki** *[ID] and **Ahmed Ghorbel** *[ID]

CODECI Laboratory, Department of Quantitative Methods, Faculty of Economics and Management (FSEG) of Sfax, University of Sfax, Sfax 3018, Tunisia
* Correspondence: amelmelki25@gmail.com (A.M.); ahmed.ghorbel@fsegs.usf.tn (A.G.);
  Tel.: +216-97249749 (A.M.)

**Abstract:** This study aims at examining whether hedging emerging Eastern Europe stock markets with commodities sectors can help in reducing market risks and whether it has the same effectiveness among different sectors. As an attempt to achieve this goal, we opt for three types of MGARCH model. These are DCC, ADCC and GO-GARCH, which are used with each bivariate series to model dynamic conditional correlations, optimal hedge ratios and hedging effectiveness. Rolling window analysis is used for out-of-sample one-step-ahead forecasts from December 1994 to June 2022. The results have shown that the commodities sectors of industrial metals and energy represent the optimal hedging instruments for emerging Eastern Europe stock markets as they have the highest hedging effectiveness. Additionally, our empirical results have proved that hedge ratios estimated by the DCC and ADCC models are very similar, which is not the case for GO-GARCH, and that hedging effectiveness is preferably estimated by the ADCC model.

**Keywords:** commodities sectors; emerging Eastern Europe stock markets; hedging effectiveness; DCC; ADCC; GO-GARCH

## 1. Introduction

The financialization of commodities (when the commodity trading volume sharply expands as a result of the increasing use of commodities as alternative assets by investors, such as pension funds, hedge funds, banks and insurance) represents the ultimate tool as part of an investment strategy against several recurrent crises. This new concept may lead to a significantly increased correlation between stock markets and commodities which, in turn, motivates researchers to shed more light and focus on the dependency between these two markets.

In this respect, to find the best strategy to hedge equity position, several studies concentrate on the link between the different types of commodities and stock markets [1–10].

The studies' results vary from positive to negative and from stable to volatile links over time. As a matter of fact, ref. [4] find significant correlations on account of higher investor interest in commodities. Ref. [10] prove increased correlations between commodities and emerging markets equities. However, refs. [7–9] calculate a negative correlation over time.

It is worth mentioning that most of the studies concentrate on the link between these two markets in the United States and other developed stock markets, while there is less interest in emerging markets. Indeed, refs. [6,11–13] confirm that the correlation trend across stock markets and commodities during financial turmoil periods, especially in the financial crisis of 2007/2008, is extremely volatile. Ref. [11] suggest that portfolio diversification across commodity and stock markets offers higher gains compared to investing only in stock markets. Furthermore, they reach the conclusion that diversification during calm periods is more efficient than in volatile times.

Consequently, investors must have a good idea about correlation movements in order to use commodities as hedging instruments or as a safe haven in times of financial turmoil

and calm periods. In this regard, ref. [14] explore the safe haven, hedging and diversification potentials of 21 commodities for 49 international stock markets. Their results reveal that gold represents a valuable instrument as a safe haven, particularly in the most developed stock markets. In addition, commodities offer the highest hedging effectiveness with advanced emerging and some developed stock markets.

It is obvious that hedging strategies vary over time. Nevertheless, it is necessary to check the time-varying relationship of returns among these two markets during different global crises, namely during the COVID-19 global pandemic and the Russo-Ukrainian war. In fact, such crises have a significant impact not only on global international demand and supply but also on stock and commodity prices.

A large number of studies have used optimal hedges and hedging effectiveness to determine the best hedging strategy. A well-known method utilized to estimate optimal hedge ratios is the portfolio-variance minimization-based approach [14–24]. This method consists of minimizing the conditional variance of a hedge.

Thereupon, several MGARCH models have been developed, such as BEKK [25], CCC (constant conditional correlation [26], DCC (dynamic conditional correlation [27] and ADCC (asymmetric dynamic conditional correlation [28]. Some models, such as the BEKK and VECH models, have been subject to criticism. One of their limitations is that the number of parameters estimated by the GARCH equation grows very fast, which limits the number of assets to two (for further details, see [29]. In order to solve this problem, CCC, DCC and ADCC are utilized to reduce the number of parameters. Yet, the problem still exists with a large number of assets. Additionally, the ADCC model is more reliable than CCC and DCC because it captures asymmetric effects. On the other hand, another MGARCH model, named the factor MGARCH model, like the OGARCH model [30] and the GO-GARCH model [31], uses a different mechanism and suggests that asset returns are produced by a set of unobserved underlying factors that are orthogonal.

This study fills a gap in the current literature. First, it provides an answer to the question of whether or not commodities sectors hedge emerging Eastern Europe stock markets at the same level of effectiveness and whether hedging effectiveness differs across sectors such as energy, precious metal, industrial metals, livestock and agriculture. Our analysis covers the major events before and after the financialization of commodity markets including recent crises such as the COVID-19 global pandemic and the Russo-Ukrainian war. Second, while several researchers apply DCC and ADCC models to compute a hedge ratio, few researchers apply the GO-GARCH model. This is why the purpose of this paper is to compare the optimal hedge ratios estimated by the DCC, ADCC and GO-GARCH models, due to the existence of a number of advantages and different mechanisms of how to interpret the information. Third, it uses rolling window analysis to forecast one-step-ahead hedge ratios produced from the three versions of GARCH model. This rigorous technique takes into consideration recent economic changes by allowing account changing variability in the data.

Our empirical research reveals that hedge ratios estimated by DCC and ADCC are very similar, which is not the case for GO-GARCH. However, the three models capture all the major turning points including the COVID-19 pandemic period and the Russian–Ukrainian war period. Hedging effectiveness varies between all alternative sectors and records the highest value in the industrial metals sector followed by the energy sector. These values are preferably estimated by the ADCC model.

This paper is made up of five main sections. The introduction, which is above, gives a brief overview of the whole study. Section two, the literature review, explores previous related academic literature. The third section, which is the methodology, supplies a comprehensive description of the corpus and data used in the study. The fourth section inquires into the results that the study has produced. The last section is dedicated to the conclusion.

## 2. Literature Review

Historically, commodities have been subject to a negative correlation with classic assets like stocks and bonds. This is what makes them an ideal asset as a hedging instrument. However, in-depth research in the literature shows that this link is positive yet uncorrelated. To fill in this gap, the present work explores research dealing with commodities as hedging instruments. In fact, a vast body of literature has been written about this matter. The aim of this section is to survey that previous research.

Silvennoinen and Thorp [4] used the DSTCC-GARCH model to estimate bivariate conditional correlation among 24 commodities futures, stocks (US, UK, Germany, France and Japan) and bonds. The results confirmed that, during a crisis, the correlations between stocks and commodities returns increase. As for [6], they confirmed that the link was extremely volatile during the financial crisis period of 2007–2008.

Some recent studies have focused on revealing the effects of the two recent crises, namely, COVID-19 and the Russo-Ukrainian war, on the dynamic connectedness between stock markets and commodities [32–35]; they tested volatility connectedness between the Indian stock market and six commodity markets using wavelet analysis. Their results showed higher volatility connectedness between the Indian stock market and all commodity markets; an increase after the COVID-19 pandemic and the transmission of contagion is significant in the medium- and long-term periods. Ref. [33] proved an average volatility contagion between oil and stock markets in the G-7 countries in addition to India and China in the COVID-19 period. Ref. [34] investigated the effect of shock transmission between US stock markets and commodities stock markets (renewable energy, oil and precious metals) in the last two crises: the COVID-19 pandemic and the Russo-Ukrainian war period. Ref. [35] examined dynamic connectedness among energy stock markets and energy commodity markets in the COVID-19 pandemic period. They proved that this crisis increased hedging effectiveness.

On the other hand, ref. [5] analyzed the diversification potential for 25 individual components for different sectors. They came to the conclusion that diversification during calm periods is more efficient than in highly volatile times. Furthermore, the commodities of energy and precious metal contribute simultaneously to reducing risks and ameliorating returns.

In addition, ref. [36] highlighted the performance gains of adding different commodity groups such as energy, precious metals, industrial metals, agriculture and livestock to a stock–bond portfolio. They proved that the best performance gains for investment strategies lie in adding aggregate commodities indices, industrial metals, precious metals and energy to a stock–bond portfolio.

As far as [37] are concerned, they evaluated dynamic correlations, portfolio weights and hedging effectiveness to study the use of commodities as hedging instruments for developed, emerging and frontier equities markets. They stated that using commodities as hedging instruments with emerging stock markets rather than developed and frontier equities markets is more efficient. What is more, ref. [38] proved that the best option is to use commodities as a hedging instrument record for emerging Eastern European stock markets. Similarly, ref. [39] used the dynamic conditional correlation (DCC) model to assess co-movements between emerging and developed markets. The results revealed that emerging markets, notably those in Asia, represent significantly lower co-movement with commodities than developed markets. The results also showed that the best opportunities for diversification occur with agricultural and precious metal commodities in less-developed markets.

Additionally, ref. [40] examined volatility spillover, optimal weights, diversification, hedge ratios and hedging effectiveness among oil and stock market sectors in Europe and the United States. They found unidirectional and bidirectional volatility spillover between oil markets and stock markets. It is worth bearing in mind that they found that hedging strategies, including oil and stocks, can reduce the portfolio risk for all sectors, especially those of basic materials from US stock markets.

Various alternative assets have been able to minimize or limit risk [4–44]. The ability of different financial assets, including gold, crude oil, VISTOXX, VIX, CDSEU and DJCOM, to cover Islamic and conventional stock markets has been compared by [41]. The best alternative asset, according to their research, is VISTOXX. Ref. [42] investigated the effectiveness of WTI, gold, VIX and five cryptocurrencies to hedge the stock market. Their research revealed that the best hedging investment differs among the various stock indices. Ref. [43] confirmed that Bitcoin and gold are able to act as hedging instruments. Ref. [44] proved that the roles of Bitcoin, VIX futures and CDS as hedging and safe havens differ across time horizons and model used.

Practically, several empirical studies have applied a number of econometric models to model asset prices and volatility dynamics. Relevant studies in this field have used GARCH models: that is to say, refs. [3,16,17,19,45]. As for [45], they used DCC, ADCC and GO-GARCH models to estimate dynamic condition correlations between oil price and equity market. Their results proved that oil price is positively correlated with equity market. Ref. [16] analyzed volatility dynamics and transmission and then compared optimal portfolio weights and optimal hedge ratios among crude oil spot and futures prices estimated by GARCH models such as CCC, VARMA-GARCH, DCC, BEKK and diagonal BEKK. After that, they suggested the best crude oil hedge strategy. First, they asserted that optimal hedge ratios (OHRs) vary over time. Second, they confirmed that the estimation of OHRs is sensitive for the approved model. Likewise, ref. [3] analyzed volatility dynamics and volatility transmission and then they compared optimal portfolio weights and optimal hedge ratios among oil and Ghanaian stock market returns using VAR-GARCH, VAR-AGARCH and DCC-GARCH models. Their findings revealed that DCC-GARCH records a high hedge effectiveness in both Ghana and Nigeria equity markets together with proving that oil assets are essential in a diversified portfolio of equities.

Ref. [17] used the A-DCC model to examine dynamic conditional correlations amongst gold and stock markets in each of the BRICS countries (Brazil, Russia, India, China and South Africa). He demonstrated that, in times of extreme stock market movement, correlations between the two assets are mostly low to negative, which means that gold can act as a safe haven. Extending the analysis to HE values and optimal portfolio weights, the results of HE indicated that hedging strategies, including the BRICS stock markets and gold assets, can reduce portfolio risk. Similar to [17,46] also studied dynamic conditional correlations between commodities (gold and oil) and the stock markets of each BRICS country. They used the DCC-FIAPARCH model to estimate optimal portfolio weights and hedge ratios. They also confirmed that oil is not an effective hedge instrument due to the BRICS economies depending on oil price changes. On the other hand, gold can serve as a hedging instrument and safe haven at the same time.

In addition, refs. [14,19–24,47] all applied three versions of GARCH models (DCC, ADCC and GO-GARCH) using a rolling window technique. This was for the purpose of estimating dynamic conditional correlation and the hedge ratios and effectiveness between stock markets and commodities. The authors unanimously agree on the fact that hedge ratios registered from the DCC and ADCC models are very similar, while those of the GO-GARCH model are different. These results are robust in terms of model refits and the number of one-step-ahead sample forecasts.

In the light of what has been surveyed in this section, it is worth asserting that most studies highlighted the hedging of developed stock markets with commodities, whereas they were not very interested in hedging emerging Eastern Europe stock markets with commodities. Moreover, they also failed to provide a balanced perspective on how stocks/commodities hedge compared to the ability of each commodity sector. In addition, these studies did not make a comparison between commodities sectors as a hedging instrument that can differ greatly from one sector to another and can be influenced by the model and refit used to estimate optimal hedge ratio. Added to that, many authors have proposed a particular selection of GARCH model (e.g., DCC-GARCH, CCC-GARCH and BEKK) and then exhibited estimated results from these models, without creating

a descriptive comparison between different models. In fact, comparing results taken from different versions of GARCH model is useful for developing our understanding of how hedge ratios differ by estimation method. Finally, many researchers have suggested estimating in-sample hedge ratios and portfolio weights. In-sample analysis is useful for comprehension of the model fit range but it does not seem to represent the best method for making a forward-looking decision. In fact, a hedge is usually preferred for capturing out-of-sample performance of hedging.

### 3. Data and Descriptive Statistics

In our empirical investigation, we used daily data over the period from December 1994 to June 2022. Such data were collected from the DataStream database. We also used the following commodities sector indices by S&P: S&P GSCI Energy Total Return Indexed, S&P GSCI Precious Metal Total Return Indexed, S&P GSCI Industrial Metals Total Return Indexed, S&P GSCI Livestock Total Return Indexed and S&P GSCI Agriculture Total Return and the following stock index: MSCI Emerging Markets Eastern Europe. We opted for this period since it is characterized by several extreme events and turbulence, including the financial crisis in Russia and Brazil (1998–1999), the internet bubble (2000), the subprime financial crisis (2007–2008), the sovereign debt crisis (2010–2012), Standard & Poor's downgrade of the United States debt rating (2011), China's stock market crisis (2015), the commodities price collapse (2015–2016), the COVID-19 global pandemic (2019–2020) and the Russo-Ukrainian war (2022). Data analysis and processing were primarily prepared using the R Studio program.

We transformed each data series into logarithmic differences in order to avoid model dependencies and to reduce heteroskedasticity. The return series are calculated as follows:

$$r_{i,t} = 100 \times log\left(\frac{P_{i,t}}{P_{i,t-1}}\right) \tag{1}$$

$P_{i,t}$ is the daily closing price at time $t$. Time series plots of daily level series for Eastern Europe, energy, precious metal, industrial metals, livestock and agriculture over the sample period are represented in Figure 1. As shown in the figure, Eastern Europe and industrial metals display similar time series patterns. There were strong downwards trends during the years 2015 and 2016, corresponding to the commodities price collapse, and towards the COVID-19 pandemic in 2020. The time series for energy and agriculture increased after the Russo-Ukrainian war in 2022 because Russia and Ukraine are the main sources of agriculture and also because Russia is the main source of energy.

Summarized statistics for returns are shown in Table 1. What is highlighted is the persistence of negative means for EM Eastern Europe returns ($-0.0142$), livestock returns ($-0.0117\%$) and agriculture returns ($-0.0072\%$). Obviously, the highest average daily returns are in the precious metal sector (0.0211%). Unconditional volatility measured by standard deviation is highest for EM Eastern Europe (2.1940), while it is lowest for the livestock sector (0.9199).

Skewness coefficients are negative for all returns, implying the existence of return distributions oriented to the left due to the recurrence of crises. This means that fat tails, which are associated with return series, are asymmetric. Furthermore, the kurtosis values of all the return series have been greater than three except for livestock and agriculture. The results indicate that return series distributions are probably skewed and leptokurtic. The JB test shows that the return distributions are abnormal. In addition to that, Ljung–Box $Q(12)$ statistics of the residuals show that the returns are autocorrelated. Moreover, the Ljung–Box of squared residuals $Q^2(12)$ presents a significant level of no serial correlation.

In this context, we examine time series graphs for squared returns in Figure 2. These time series plots exemplify the existence of an ARCH effect. On this basis, the use of the DCC, ADCC and GO-GARCH models is perfectly suitable for establishing dynamic dependence among the Emerging Eastern Europe stock markets and commodities sectors.

For this purpose, the choice of a hedging instrument for emerging Eastern Europe stock markets with alternative commodities sectors depends on the degree of correlation.

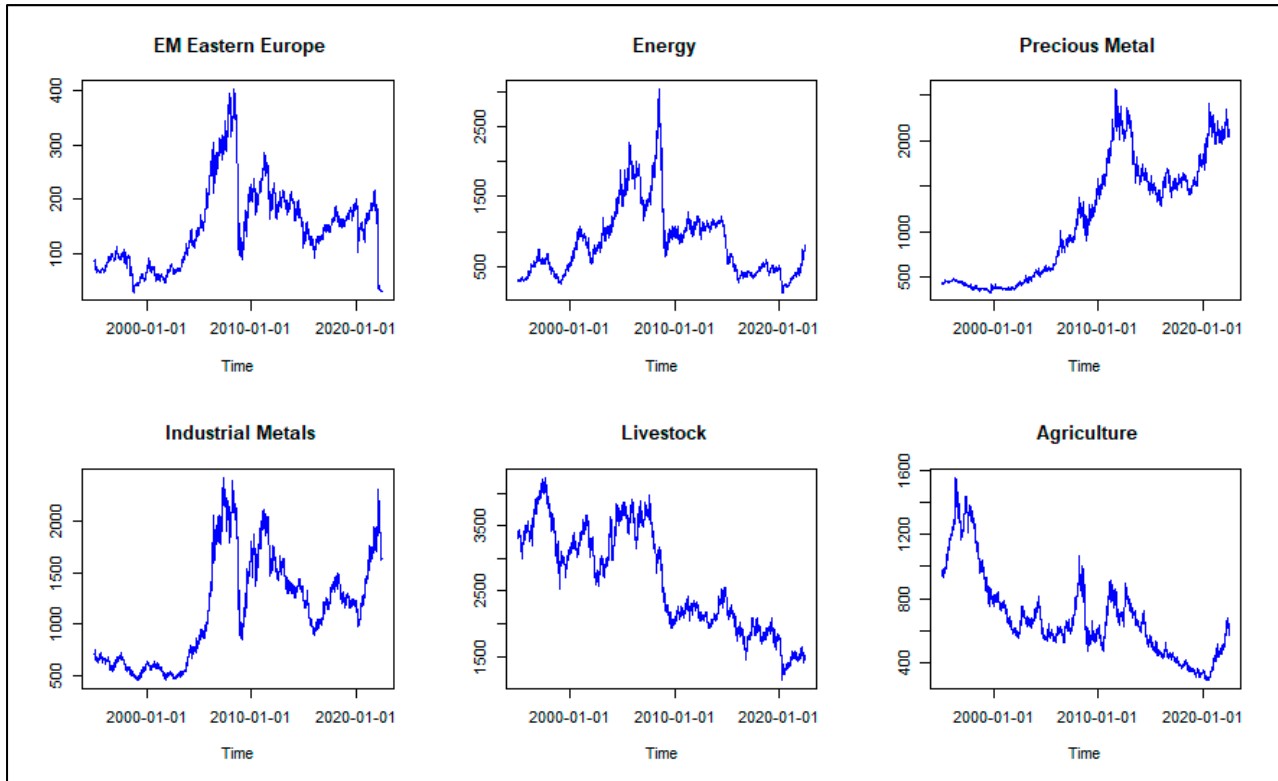

**Figure 1.** Time series plots.

**Table 1.** Descriptive statistics.

| | EM Eastern Europe | Energy | Precious Metal | Industrial Metals | Livestock | Agriculture |
|---|---|---|---|---|---|---|
| Mean | −0.0142 | 0.0117 | 0.0211 | 0.0115 | −0.0117 | −0.0072 |
| Median | 0.0783 | 0.0164 | 0.0211 | 0.0115 | −0.0117 | 0.0000 |
| Max | 19.1190 | 15.9825 | 8.7625 | 7.5883 | 5.3018 | 7.1568 |
| Min | −80.6484 | −30.1688 | −10.1046 | −9.0150 | −6.2366 | −7.4752 |
| Std. dev. | 2.1940 | 2.0574 | 1.0808 | 1.2794 | 0.9199 | 1.1658 |
| Skewness | −8.0904 | −0.7955 | −0.2730 | −0.2323 | −0.2346 | −0.0655 |
| Kurtosis | 273.5993 | 12.9070 | 6.9961 | 3.4425 | 1.9837 | 2.7906 |
| JB test | 22,463,660 *** | 50,582 *** | 14,730 *** | 3610.2 *** | 1243.6 *** | 2335.4 *** |
| $Q(12)$ | 214.09 *** | 27.967 ** | 30.889 *** | 22.546 ** | 39.864 *** | 11.921 |
| $Q^2(12)$ | 413.18 *** | 1395.3 *** | 781.27 *** | 2988.3 *** | 4617.3 *** | 1706 *** |

Note: *** and ** indicates rejection of the null hypothesis at the 1% and 5% levels, respectively.

We move now to analyze the correlations between squared daily returns in Table 2. To check interdependence between variables, we also calculated an unconditional correlation matrix (Table 3). We found that unconditional correlations between precious metal and industrial metals are high, followed by those of EM Eastern Europe and industrial metals. However, this unconditional correlation is low as far as precious metals and livestock are concerned.

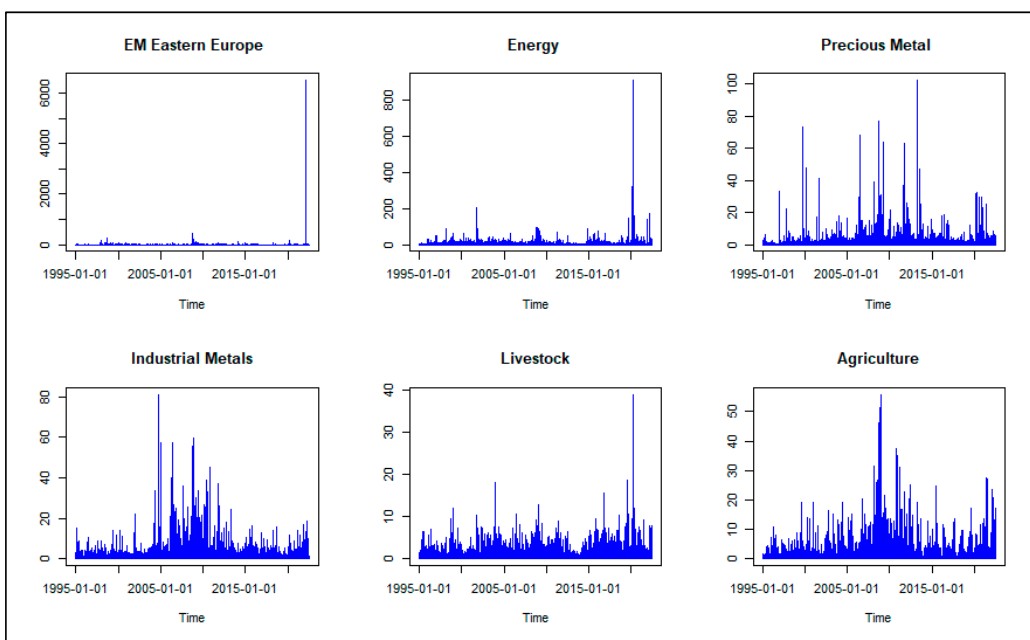

**Figure 2.** Squared daily returns.

**Table 2.** Correlations between daily returns.

|  | EM Eastern Europe | Energy | Precious Metal | Industrial Metals | Livestock | Agriculture |
|---|---|---|---|---|---|---|
| EM Eastern Europe | 1.0000 | 0.2645 | 0.1427 | 0.2976 | 0.1026 | 0.1653 |
| Energy | 0.2645 | 1.0000 | 0.1975 | 0.2932 | 0.1149 | 0.2641 |
| Precious metal | 0.1427 | 0.1975 | 1.0000 | 0.3144 | 0.0384 | 0.2169 |
| Industrial metals | 0.2976 | 0.2932 | 0.3144 | 1.0000 | 0.1153 | 0.2702 |
| Livestock | 0.1026 | 0.1149 | 0.0384 | 0.1153 | 1.0000 | 0.1195 |
| Agriculture | 0.1653 | 0.2641 | 0.2169 | 0.2702 | 0.1195 | 1.0000 |

**Table 3.** DCC parameter estimates.

|  | Eastern European Markets and Energy | | Eastern European Markets and Precious Metal | | Eastern European Markets and Industrial Metals | | Eastern European Markets and Livestock | | Eastern European Markets and Agriculture | |
|---|---|---|---|---|---|---|---|---|---|---|
|  | Coeff | *p*-Value | Coeff | *p*-Value | Coeff | *p*-Value | Coeff | *p*-Value | Coeff | *p*-Value |
| $\mu_1$ | 0.0588 | 0.0731 | 0.0588 | 0.0731 | 0.0588 | 0.0731 | 0.0588 | 0.0730 | 0.0588 | 0.0733 |
| $a_1$ | 0.1584 | 0.0000 | 0.1584 | 0.0000 | 0.1584 | 0.0000 | 0.1584 | 0.0000 | 0.1584 | 0.0000 |
| $\omega_1$ | 0.0270 | 0.2999 | 0.0270 | 0.3000 | 0.0270 | 0.3002 | 0.0270 | 0.3004 | 0.0270 | 0.2999 |
| $\alpha_1$ | 0.1195 | 0.0858 | 0.1195 | 0.0859 | 0.1195 | 0.0862 | 0.1195 | 0.0863 | 0.1195 | 0.0857 |
| $\beta_1$ | 0.8794 | 0.0000 | 0.8794 | 0.0000 | 0.8794 | 0.0000 | 0.8794 | 0.0000 | 0.8794 | 0.0000 |
| $\lambda_1$ | 5.4784 | 0.0000 | 5.4784 | 0.0000 | 5.4784 | 0.0000 | 5.4784 | 0.0000 | 5.4784 | 0.0000 |
| $\mu_2$ | 0.0812 | 0.0483 | −0.0183 | 0.1412 | 0.0062 | 0.7878 | 0.0001 | 0.9920 | −0.0132 | 0.5547 |
| $a_2$ | 0.0173 | 0.4972 | −0.0262 | 0.3017 | 0.0567 | 0.0324 | 0.0071 | 0.7887 | 0.0389 | 0.1111 |
| $\omega_2$ | 0.0230 | 0.0278 | 0.0057 | 0.1442 | 0.0663 | 0.0051 | 0.0123 | 0.0449 | 0.0208 | 0.0142 |
| $\alpha_2$ | 0.0359 | 0.0000 | 0.0545 | 0.0202 | 0.0488 | 0.0020 | 0.0508 | 0.0000 | 0.0528 | 0.0000 |
| $\beta_2$ | 0.9584 | 0.0000 | 0.9406 | 0.0000 | 0.8727 | 0.0000 | 0.9303 | 0.0000 | 0.9225 | 0.0000 |
| $\lambda_2$ | 6.3685 | 0.0000 | 3.4819 | 0.0000 | 8.5866 | 0.0000 | 20.5696 | 0.0426 | 12.7332 | 0.0008 |
| $\theta_1$ | 0.0000 | 0.9996 | 0.0041 | 0.5071 | 0.0106 | 0.3059 | 0.0028 | 0.4914 | 0.0180 | 0.2277 |
| $\theta_2$ | 0.9280 | 0.0076 | 0.9594 | 0.0000 | 0.9645 | 0.0000 | 0.9843 | 0.0000 | 0.8385 | 0.0000 |
| $Q_1(12)$ | 22.031 | 0.0371 | 21.793 | 0.0399 | 21.953 | 0.0380 | 22.051 | 0.0369 | 21.807 | 0.0397 |
| $Q_2(12)$ | 13.331 | 0.3454 | 15.058 | 0.2383 | 27.538 | 0.0064 | 26.742 | 0.0084 | 14.061 | 0.2969 |
| $Q_1^2(12)$ | 14.778 | 0.2538 | 14.289 | 0.2827 | 14.486 | 0.2707 | 14.82 | 0.2514 | 15.004 | 0.2412 |
| $Q_2^2(12)$ | 22.874 | 0.0288 | 7.1733 | 0.846 | 14.055 | 0.2972 | 13.531 | 0.3316 | 16.055 | 0.1887 |
| AIC | 7.4461 | | 5.5549 | | 6.1970 | | 5.9435 | | 6.1650 | |
| BIC | 7.5028 | | 5.6115 | | 6.2537 | | 6.0002 | | 6.2217 | |
| Shibata | 7.4459 | | 5.5546 | | 6.1968 | | 5.9433 | | 6.1648 | |
| Max likelihood | −5568.566 | | −4150.149 | | −4631.785 | | −4441.614 | | −4607.785 | |

Note: Estimation of the parameters of the ARMA(1,0)–DCC(1,1) model with residuals follows the bivariate t-student distribution for each pair.

## 4. Methodology

In this paper, we use the dynamic conditional correlation model of [27], the asymmetric dynamic conditional correlation model of [28] and the generalized orthogonal GARCH model of [31] to estimate volatility dynamics, conditional correlations bivariate and hedge ratios among the Eastern European markets/energy sector, Eastern European markets/precious metal sector, Eastern European markets/industrial metals sector, Eastern European markets/livestock sector and Eastern European markets/agriculture sector.

Let $(r_t)$ be an $(n \times 1)$ vector of asset returns. An auto-regressive (AR) (1) process for $(r_t)$ conditional on information set $I_{t-1}$ can be expressed as:

$$r_t = \mu + a r_{t-1} + \varepsilon_t \qquad (2)$$

The residuals in Equation (2) are calculated as follows:

$$\varepsilon_t = H_t^{1/2} Z_t \qquad (3)$$

where $H_t$ is an $n \times n$ conditional variance–covariance matrix of $r_t$ and $Z_t$ is an $n \times 1$ identically and independently distributed (i.i.d) random vector of residuals.

The [27] dynamic conditional correlation (DCC) model is estimated in two steps. In the first step, GARCH parameters are estimated, and, as a second step, conditional correlations are estimated.

$$H_t = D_t R_t D_t \qquad (4)$$

The diagonal matrix of assets $(D_t)$ has time-varying standard deviations. It is modeled as:

$$D_t = diag\left(h_{1,t}^{1/2}, \ldots h_{n,t}^{1/2}\right) \qquad (5)$$

where $(R_t)$ represents the time-varying conditional correlation matrix of assets. It is written as follows:

$$R_t = diag\left(q_{1,t}^{-1/2}, \ldots q_{n,t}^{-1/2}\right) Q_t \, diag\left(q_{1,t}^{-1/2}, \ldots q_{n,t}^{-1/2}\right) \qquad (6)$$

The univariate GARCH(1,1) is used to write an expression for $h_{i,t}$ in Equation (5). It is stated as:

$$h_{i,t} = \omega_i + \alpha_i \varepsilon_{i,t-1}^2 + \beta_i h_{i,t-1} \qquad (7)$$

The symmetric positive matrix $(Q_t)$ of elements $q_{ij;t}$ is:

$$Q_t = (1 - \theta_1 - \theta_2)\overline{Q} + \theta_t Z_{t-1} Z'_{t-1} + \theta_2 Q_{t-1} \qquad (8)$$

$\overline{Q}$ illustrates the $n \times n$ unconditional correlation matrix of standardized residuals $Z_{i,t}\left(Z_{i,j} = \varepsilon_{i,t}/\sqrt{h_{i,t}}\right)$. Positive parameters $\theta_1$ and $\theta_2$ associated with the exponential smoothing process are utilized to capture the effects of previous shocks and previous dynamic conditional correlations on current dynamic conditional correlations. If $\theta_1 + \theta_2 < 1$, the DCC model is mean-reverting, and if $\theta_1 + \theta_2 > 1$, then the DCC model is integrated. The correlation estimate is as follows:

$$\rho_{i,j,t} = \frac{q_{i,j,t}}{\sqrt{q_{i,i,t} q_{j,j,t}}} \qquad (9)$$

To solve the problem of asymmetry effects of the DCC model, ref. [28] build on ADCC so that:

$$h_{i,t} = \omega_i + \alpha_i \varepsilon_{i,t-1}^2 + \beta_i h_{i,t-1} + d_i \varepsilon_{i,t-1}^2 I(\varepsilon_{i,t-1}) \qquad (10)$$

where there is an indicator function $I(\varepsilon_{i,t-1})$. According to the ADCC model, the dynamics of $Q$ can be specified as the following:

$$Q_t = \left(\overline{Q} - A'\overline{Q}A - B'\overline{Q}B - G'\overline{Q}^- G\right) + A'Z_{t-1}Z'_{t-1}A + B'Q_{t-1}B + G'Z_t^- Z'^-_{t-1}G \qquad (11)$$

In Equation (11), $A$, $B$ and $G$ are n × n parameter matrices. The zero-threshold standardized errors vector $Z_t^-$ is equal to $Z_t$ when less than 0 and 0 otherwise. $\overline{Q}$ and $\overline{Q}^-$ represent the unconditional matrices of $Z_t$ and $Z_t^-$, respectively.

The GO-GARCH model of [31] supposes $r_t$, conditional mean ($m_t$) and error term ($\varepsilon_t$):

$$r_t = m_t + \varepsilon_t \tag{12}$$

A set of unobservable independent factors $f_t$ is employed for GO-GARCH model maps. The $r_t - m_t$ return is:

$$\varepsilon_t = A f_t \tag{13}$$

In the above equation, $A$ is a mixing matrix that can be decomposed into an orthogonal matrix $U$ and unconditional covariance matrix $\Sigma$.

$$A = \sum^{1/2} U \tag{14}$$

The rows in matrix $A$ represent assets, while the columns refer to factors ($f$) and are modeled as follows:

$$f_t = H_t^{1/2} Z_t \tag{15}$$

The random variable $Z_t$ affords the characteristics $E(Z_{i,t}) = 0$ and $E\left(Z_{i,t}^2\right) = 1$. Factor conditional variances $h_{it}$ can be displayed in terms of a GARCH process. The unconditional distribution of the factors, $f$, satisfies $E(f_t) = 0$ and $E\left(f_i f_t'\right) = I$. Combining Equations (12)–(14), for the conditional mean, the equation becomes:

$$r_t = m_t + A H_t^{1/2} Z_t \tag{16}$$

The conditional covariance matrix of the returns ($r_t - m_t$) can be written as:

$$\sum\nolimits_t = A H_t A' \tag{17}$$

Therefore, the GO-GARCH model introduced by [31] is claimed to be based on two assumptions. First, the mixing matrix ($A$) is assumed to be time-invariant. Second, $H_t$ constitutes the diagonal matrix as for the DCC and ADCC models. On the other hand, the OGARCH model is a special case of the GO-GARCH model whereby the mixing matrix ($A$) is suggested to be orthogonal. In this study, we use independent component analysis (ICA) [12,48] to estimate matrix U.

## 5. Empirical Results

Econometric tests of the model construction strategy validate the application of the GARCH(1,1) variance equation and AR (1) term in the mean equation for each model: that is to say, DCC, ADCC and GO-GARCH. In order to account for an abnormal, fat and left-oriented distribution of the returns, DCC and ADCC are modeled with a multivariate student distribution (MVT). GO-GARCH is modeled with a multivariate affine negative inverse Gaussian (MANIG) distribution. We use a multivariate GARCH model with each bivariate series to avoid the likelihood function flattening.

### 5.1. Regression Results

DCC and ADCC parameter estimates are presented in Tables 3 and 4. Our empirical results reveal that the estimated coefficients of short-term persistence ($\alpha$) are positive and statistically significant in each case. The estimated coefficient and long-term persistence volatility ($\beta$) ARCH effects are statistically significant for each variable. They indicate the persistence of alternate volatility packets for all variables. We also witness that short-term persistence ($\alpha$) is less than long-term persistence ($\beta$) and the sum ($\alpha + \beta$) is less than unity for all sample variables. The estimated asymmetric term ($\gamma$) helps to verify the absence or presence of the leverage effect. The parameters ($\gamma$) are positive and statistically significant

for industrial metals, livestock and Eastern Europe. This means that, as concerns industrial metals, livestock and Eastern Europe, negative shocks tend to increase conditional variance more than positive shocks of the same magnitude. The results confirm the absence of leverage effects. The estimated asymmetrical term is negative regarding energy, precious metal and agriculture, indicating that negative residuals tend to decrease conditional volatility.

**Table 4.** ADCC parameter estimates.

| | Eastern European Markets and Energy | | Eastern European Markets and Precious Metal | | Eastern European Markets and Industrial Metals | | Eastern European Markets and Livestock | | Eastern European Markets and Agriculture | |
|---|---|---|---|---|---|---|---|---|---|---|
| | Coeff | *p*-Value | Coeff | *p*-Value | Coeff | *p*-Value | Coeff | *p*-Value | Coeff | *p*-Value |
| $\mu_1$ | 0.0336 | 0.2958 | 0.0336 | 0.2950 | 0.0336 | 0.2952 | 0.0336 | 0.2952 | 0.0336 | 0.2955 |
| $a_1$ | 0.1635 | 0.0000 | 0.1635 | 0.0000 | 0.1635 | 0.0000 | 0.1635 | 0.0000 | 0.1635 | 0.0000 |
| $\omega_1$ | 0.0210 | 0.2125 | 0.0210 | 0.2111 | 0.0210 | 0.2115 | 0.0210 | 0.2113 | 0.0210 | 0.2120 |
| $\alpha_1$ | 0.1283 | 0.0162 | 0.1283 | 0.0160 | 0.1283 | 0.0160 | 0.1283 | 0.0159 | 0.1283 | 0.0161 |
| $\beta_1$ | 0.8930 | 0.0000 | 0.8930 | 0.0000 | 0.8930 | 0.0000 | 0.8930 | 0.0000 | 0.8930 | 0.0000 |
| $\gamma_1$ | 0.1409 | 0.0825 | 0.1409 | 0.0821 | 0.1409 | 0.0824 | 0.1409 | 0.0827 | 0.1409 | 0.0829 |
| $\lambda_1$ | 5.3018 | 0.0000 | 5.3018 | 0.0000 | 5.3018 | 0.0000 | 5.3018 | 0.0000 | 5.3018 | 0.0000 |
| $\mu_2$ | 0.0791 | 0.0573 | −0.0106 | 0.3640 | −0.0095 | 0.6943 | −0.0186 | 0.3654 | −0.0093 | 0.6316 |
| $a_2$ | 0.0167 | 0.5266 | −0.0271 | 0.1756 | 0.0607 | 0.0235 | 0.0136 | 0.6084 | 0.0429 | 0.0068 |
| $\omega_2$ | 0.0144 | 0.0095 | 0.0089 | 0.1281 | 0.0851 | 0.0052 | 0.0084 | 0.0000 | 0.0240 | 0.0178 |
| $\alpha_2$ | 0.0374 | 0.0000 | 0.0741 | 0.0004 | 0.0417 | 0.0087 | 0.0303 | 0.0000 | 0.0586 | 0.0000 |
| $\beta_2$ | 0.9643 | 0.0000 | 0.9377 | 0.0000 | 0.8749 | 0.0000 | 0.9662 | 0.0000 | 0.9280 | 0.0000 |
| $\gamma_2$ | −0.1617 | 0.3015 | −0.3723 | 0.0078 | 0.8453 | 0.0276 | 1.0000 | 0.0000 | −0.2121 | 0.1223 |
| $\lambda_2$ | 6.3895 | 0.0000 | 3.5581 | 0.0000 | 9.3481 | 0.0000 | 99.9996 | 0.2810 | 13.0833 | 0.0012 |
| $\theta_1$ | 0.0000 | 0.9999 | 0.0040 | 0.5496 | 0.0106 | 0.4527 | 0.0030 | 0.6642 | 0.0116 | 0.6585 |
| $\theta_2$ | 0.9340 | 0.0000 | 0.9624 | 0.0000 | 0.9537 | 0.0000 | 0.9850 | 0.0000 | 0.8678 | 0.0000 |
| $\theta_3$ | 0.0000 | 0.9999 | 0.0000 | 0.9999 | 0.0000 | 0.9999 | 0.0000 | 0.9998 | 0.0117 | 0.5543 |
| $Q_1(12)$ | 23.665 | 0.0225 | 23.446 | 0.0241 | 23.558 | 0.0233 | 23.635 | 0.0228 | 23.276 | 0.0254 |
| $Q_2(12)$ | 12.191 | 0.4305 | 16.671 | 0.1624 | 25.365 | 0.0131 | 28.277 | 0.0050 | 13.893 | 0.3076 |
| $Q_1^2(12)$ | 24.957 | 0.0150 | 24.556 | 0.0170 | 24.758 | 0.0160 | 25.026 | 0.0147 | 25.176 | 0.0140 |
| $Q_2^2(12)$ | | | 23.58 | 0.0231 | 11.598 | 0.4785 | 15.47 | | 14.57 | 0.2658 |
| AIC | 7.4544 | | 5.5573 | | 6.2010 | | 5.9395 | | 6.1702 | |
| BIC | 7.5217 | | 5.6246 | | 6.2683 | | 6.0069 | | 6.2375 | |
| Shibata | 7.4540 | | 5.5570 | | 6.2007 | | 5.9392 | | 6.1699 | |
| Max likelihood | −5571.772 | | −4148.99 | | −4631.762 | | −4435.662 | | −4608.664 | |

Note: Estimation of the parameters of the ARMA(1,0)–ADCC(1,1) model with residuals follows the bivariate t-student distribution for each pair.

The shape parameters ($\lambda$) are equal to the degrees of freedom in the t distribution. Indeed, a higher ($\lambda$) number indicates that the t-distribution curve approaches a normal distribution. Moreover, the industrial metals, livestock and agriculture sector indices have the highest shape parameters (over seven). These results suggest that the distributions of other variables have heavier tails compared to those of industrial metals, livestock and agriculture from the DCC and ADCC estimated values of ($\lambda$) (less than seven) for energy, precious metal and Eastern Europe. Parameters $\theta_1$ and $\theta_1$ are each non-negative and statistically significant. Their sum ($\theta_1 + \theta_2$) is less than one, establishing the presence of adjustment in dynamic pair-wise correlations.

GO-GARCH parameter estimates are presented in Table 5. An estimation of the GO-GARCH model is presented in three empirical results and reported in three tables. The first table exhibits the rotation matrix ($U$), the second table displays the mixing matrix ($A$) and the third table shows the estimated parameters of the GO-GARCH model. In addition, the rotation matrix $U$ is orthogonal because $U^T U = I$. In this table, the ($F_i$) factor presents the weight ($\omega$) and the short-term ($\alpha$) and long-term ($\beta$) ARCH effects estimated by the shape and skewness of the distributions. For each factor, the short-term ARCH effects ($\alpha$) appear to be less than the long-term ARCH effects ($\beta$) and their sum is less than one. This indicates a mean reversion process for different volatilities. Indeed, these results are similar to those demonstrated by the DCC and ADCC models.

**Table 5.** GO-GARCH parameter estimates.

| | Eastern European Markets and Energy | | Eastern European Markets and Precious Metal | | Eastern European Markets and Industrial Metals | | Eastern European Markets and Livestock | | Eastern European Markets and Agriculture | |
|---|---|---|---|---|---|---|---|---|---|---|
| **Rotation matrix U** | | | | | | | | | | |
| | U(1) | U(2) | U(1) | U(2) | U(1) | U(2) | U(1) | U(2) | U(1) | U(2) |
| U(1) | −0.6037 | −0.7971 | −0.9920 | 0.1258 | −0.3722 | −0.92812 | −0.0865 | −0.9963 | −0.9879 | 0.1545 |
| U(2) | −0.7971 | 0.6037 | 0.1258 | 0.9920 | −0.92812 | 0.3722 | −0.9963 | 0.0865 | 0.1545 | 0.9879 |
| **Mixing matrix A** | | | | | | | | | | |
| | A(1) | A(2) | A(1) | A(2) | A(1) | A(2) | A(1) | A(2) | A(1) | A(2) |
| A(1) | 0.2451 | 2.1799 | 2.1864 | −0.1790 | 0.5418 | 2.1257 | 0.1426 | 2.1890 | 2.1842 | −0.2038 |
| A(2) | 2.0315 | 0.3215 | 0.0659 | −1.0787 | 1.2764 | 0.0715 | 0.9187 | 0.0340 | 0.0839 | −1.1625 |
| **GO-GARCH parameter estimates** | | | | | | | | | | |
| | F1 | F2 | F1 | F2 | F1 | F2 | F1 | F2 | F1 | F2 |
| $\omega$ | 0.0065 | 0.0099 | 0.0099 | 0.0026 | 0.0052 | 0.0121 | 0.0120 | 0.0096 | 0.0098 | 0.0091 |
| $\alpha$ | 0.0609 | 0.1032 | 0.1077 | 0.0375 | 0.0381 | 0.1086 | 0.0478 | 0.1029 | 0.1040 | 0.0520 |
| $\beta$ | 0.9345 | 0.8847 | 0.8808 | 0.9614 | 0.9563 | 0.8753 | 0.9401 | 0.8856 | 0.8845 | 0.9392 |
| Skew | −0.0880 | −0.1560 | −0.1572 | 0.0355 | −0.0490 | −0.1393 | −0.0887 | −0.1652 | −0.1671 | −0.0447 |
| Shape | 1.6671 | 1.6516 | 1.6462 | 0.8045 | 2.4167 | 1.7575 | 4.5836 | 1.7182 | 1.6809 | 2.6277 |

*5.2. Dynamic Conditional Correlations*

In accordance with [18,20,21] we used rolling window analysis to estimate one-step-ahead dynamic conditional correlation and conditional variance–covariance for each day to produce a new optimal hedge ratio. Firstly, we collected 7174 observations and then fixed the estimation window at every h (500, 1500 or 2000) to produce $(7174 - h)$ one step ahead. Secondly, we refitted each GARCH model for every 20, 40 and 60 daily observations. Our purpose was to evaluate the sensitivity of the hedging effectiveness indicator regarding the different versions of MGARCH models, different forecast lengths and different refits used to choose among short- and long-run effectiveness of commodities sectors, with the objective of hedging Eastern European stock markets.

Rolling one-step-ahead dynamic conditional correlations between Eastern European markets and a corresponding position and energy, precious metal, industrial metals, livestock and agriculture are displayed in Figure 3.

In general, the correlations from DCC and ADCC have very similar dynamics. However, the correlations between DCC and GO-GARCH models, and ADCC and GO-GARCH models, show a different pattern. These results are in line with those obtained by [18,20,21]. In addition, the correlations from the GO-GARCH model are more volatile than for the DCC and ADCC models. Recently, the dynamic correlations estimated from the three models have increased since late 2020 with the emergence of the COVID-19 pandemic and the Russo-Ukrainian war of 2022. Furthermore, although dynamic conditional correlations appear to fluctuate between negative and positive, they tend to be more positive. Therefore, it is necessary to change the hedging strategy over time.

Table 6 summarizes the correlations between correlations. It is worth noting that the results of all the pairs of correlations prove that the dynamic conditional correlations estimated by the DCC and ADCC models are very high. This is not the case for the pairs of correlations estimated by the DCC and GO-GARCH models or the ADCC and GO-GARCH models, where they are considerably lower.

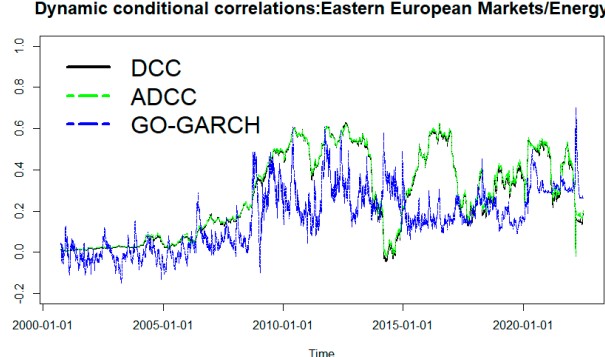

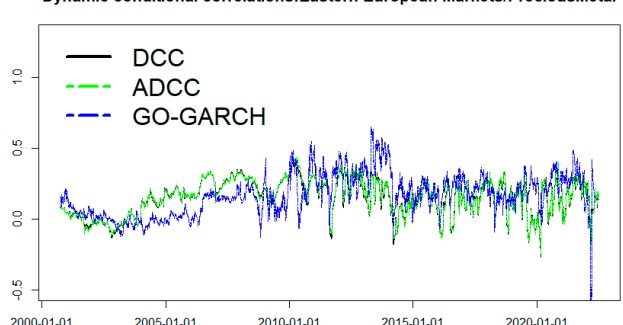

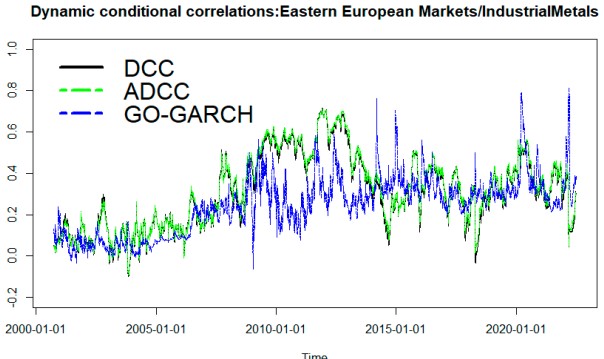

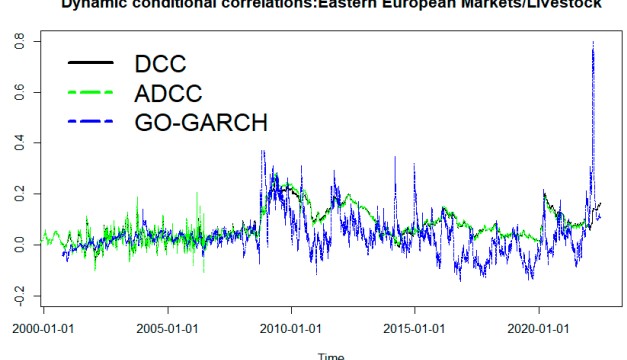

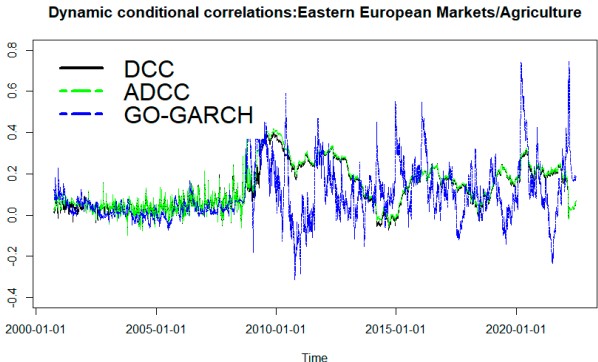

**Figure 3.** Rolling one-step-ahead dynamic conditional correlations.

**Table 6.** Correlations between correlations (window 1500/refit = 20).

|  | Eastern European Markets and Energy | Eastern European Markets and Precious Metal | Eastern European Markets and Industrial Metals | Eastern European Markets and Livestock | Eastern European Markets and Agriculture |
|---|---|---|---|---|---|
| DCC/ADCC | 0.9990 | 0.9965 | 0.9978 | 0.9908 | 0.9921 |
| DCC/GO-GARCH | 0.6290 | 0.46571 | 0.5859 | 0.3113 | 0.6417 |
| ADCC/GO-GARCH | 0.6417 | 0.4664 | 0.6013 | 0.4585 | 0.3324 |

*5.3. Hedging and Risk Management*

In theory, several models can be used to estimate the optimal hedge ratios computed for optimal hedging effectiveness. The well-known models applied are those that minimize portfolio variance, known as minimum-variance hedge-ratio models. In order to reduce the risk of investing in emerging Eastern Europe stock markets, DCC, ADCC and GO-GARCH models are used to estimate optimal hedge ratios. Subsequently, efficient hedging strategies are identified.

Returns on the emerging Eastern Europe stock markets and commodity sectors can be represented as follows:

$$R_{p,t} = R_{S,t} - \gamma_t R_{C,t} \tag{18}$$

where $R_{p,t}$ is the return on the hedged portfolio and $R_{S,t}$ and $R_{C,t}$ denote, respectively, the returns on emerging Eastern Europe stock markets and commodity sectors. Parameter $\gamma_t$ represents the dynamic hedge ratio. This parameter implies that if an investor decides to take a long position in the emerging Eastern Europe stock markets index, then the hedge ratio will be the sectors commodities index to be sold. The return variance of the conditional hedged portfolio on information set at time $t-1$ is:

$$var(R_{p,t}, I_{t-1}) = var(R_{S,t}, I_{t-1}) - 2\gamma_t cov(R_{C,t}, R_{S,t}I_{t-1}) + \gamma_t^2 var(R_{C,t}, I_{t-1}) \tag{19}$$

where $\gamma_t$ indicates the optimal hedge ratios (OHRs) that minimize the conditional variance of the hedged portfolio conditional on the information set $I_{t-1}$, while the OHR can be obtained by taking the partial derivative of the variance equal to zero [49], which is expressed as the following:

$$\gamma_t^* I_{t-1} = \frac{cov(R_{S,t}, R_{C,t} \backslash I_{t-1})}{var(R_C \backslash I_{t-1})} \tag{20}$$

A variance–covariance matrix estimated from MGARCH models can be used to construct hedge ratios [50]. In the $(\gamma_t^*)$ formula, if $(\gamma_t^*)$ is positive, the investor should take a long position in a first asset $(i)$ at time t and a short position in a second asset $(j)$ at time $t$. An expression of the hedge ratio is given by:

$$\gamma_t^* \backslash I_{t-1} = h_{SC,t}/h_{C,t} \tag{21}$$

where $h_{SC,t}$ designates the conditional covariance between emerging Eastern Europe stock markets and commodity sectors returns and $h_{C,t}$ designates the conditional variance of commodity sectors returns. Following [16,51] the hedging effectiveness $(HE)$ index is:

$$HE = \frac{var_{unhedged} - var_{hedged}}{var_{unhedged}} \tag{22}$$

The hedging effectiveness $(HE)$ is able to give an idea on how to assess the performance of dynamic hedge ratios usefully and, therefore, reduce portfolio risk. While $var_{unhedged}$ indicates the variance of returns on an unhedged stock portfolio, $var_{unhedged}$ indicates the variance of returns on a hedged stock portfolio with alternative assets (e.g., emerging Eastern Europe Stock markets and commodities sectors). A higher $HE$ indicator represents a greater risk of reduction. A hedge strategy is perfect when the $HE$ value is near to one. Following [18,20,21] we use rolling window analysis to construct out-of-sample hedge ratios. For period t, we forecast one-period-ahead conditional volatility, and these forecasts are used to make a one-period-ahead hedge ratio. Some 7174 daily observations are employed to produce 1500 one-period-ahead 5674 conditional variance/covariance matrices based on the application of DCC, ADCC and GO-GARCH models. Then, these forecasted variance/covariance matrices are refit every 20 observations to construct hedge ratios. Optimal hedge ratios estimated between Eastern European markets and a corresponding position of energy, precious metal, industrial metals, livestock and agriculture are displayed in Figure 4.

Notice that, for most of the sample periods, all commodity sectors have a positive optimal hedge ratio and high variability with Eastern European markets. Therefore, investors should change their investment strategies. Hedge ratios obtained by DCC and ADCC have very similar patterns, whereas they are very different from the GO-GARCH model. Also notice that the volatility of these results conforms with events related to the global economic system, such as the subprime financial crisis in 2007–2008, the European

debt crisis in 2010–2011, Standard & Poor's downgrade of the US credit rating from AAA to AA+ in 2011, China's stock market crisis in 2015, the COVID-19 global pandemic in 2019 and the Russo-Ukrainian war.

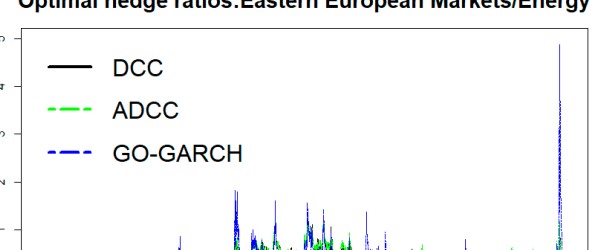

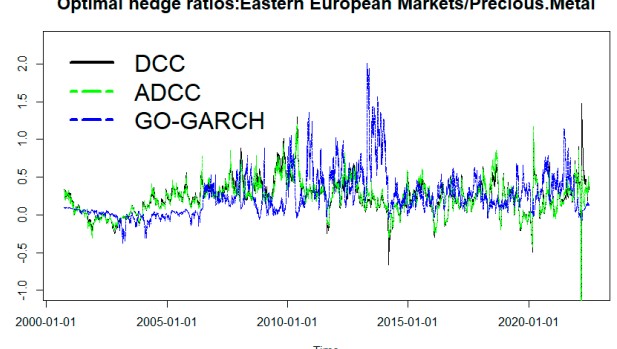

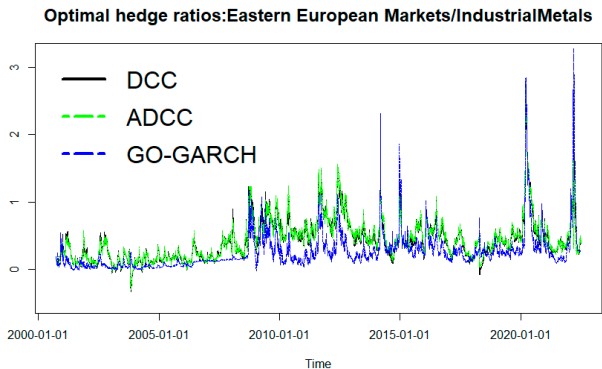

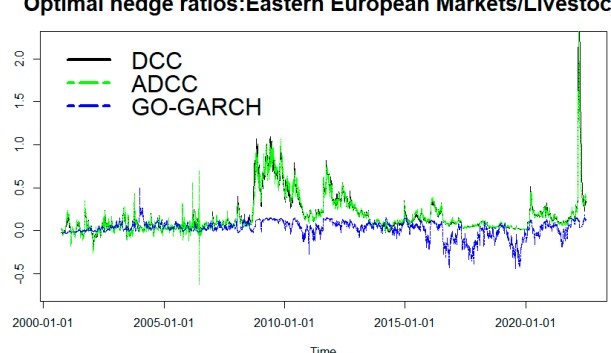

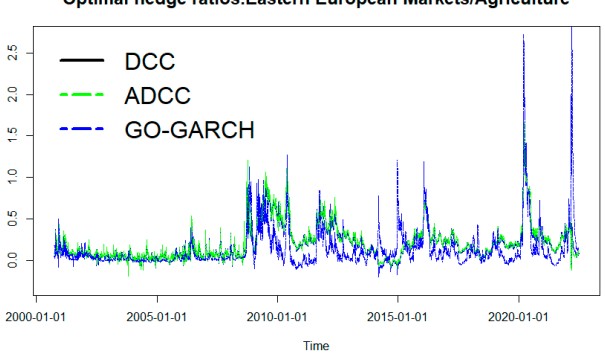

**Figure 4.** Rolling one-step-ahead optimal hedge ratios.

Table 7 displays the correlations between hedge ratios. The correlations between the hedge ratios produced from DCC and ADCC are highly correlated compared to the correlations between DCC/GO-GARCH and ADCC/GO-GARCH, which is consistent with Figure 4.

**Table 7.** Correlations between hedge ratios (window 1500/refit = 20).

|  | Eastern European Markets and Energy | Eastern European Markets and Precious Metal | Eastern European Markets and Industrial Metals | Eastern European Markets and Livestock | Eastern European Markets and Agriculture |
|---|---|---|---|---|---|
| DCC/ADCC | 0.9900 | 0.9583 | 0.9849 | 0.9593 | 0.9836 |
| DCC/GO-GARCH | 0.6535 | 0.1332 | 0.8194 | 0.3626 | 0.6421 |
| ADCC/GO-GARCH | 0.6371 | 0.1908 | 0.7799 | 0.4036 | 0.6401 |

Tables 8–10 show the summarized statistics of hedging effectiveness and hedge ratios among each pair. These tables show that the means for the hedge ratios are positive for all alternative assets. The mean hedge-ratio pairs between Eastern European markets and energy are 0.2167 for the DCC model (Table 8; refit = 20). We interpret these values as a USD 1 long position in Eastern European markets that can be hedged on average for USD 0.2167 with a short position in energy. By comparison, the mean values of the Eastern European markets/energy hedge ratio are USD 0.2257 and USD 0.1824 with respect to the ADCC and GO-GARCH models. Table 11 demonstrates the top hedging alternatives according to the indicator ($HE$) relevant to the different refits and windows. In our study, we compare all the alternatives based on the observation of hedging effectiveness ($HE$) between Eastern European markets and commodities sectors. First, we can see that the values of ($HE$) are positive for all alternative assets. Second, the highest values of ($HE$) sectors are seen in the industrial metals sector at 14.42% (Table 10; refit = 20) followed by the energy sector at 13.30% (Table 10; refit = 20). Finally, the lowest value of ($HE$) is recorded in the livestock sector at 0.74% (Table 8; refit = 60). Our experimental results show that investors can hedge Eastern European markets more efficiently by introducing industrial metals indices into their portfolios, while livestock indices are a very weak instrument to hedge Eastern European markets.

**Table 8.** Hedge-ratio summary statistics and hedging effectiveness (HE) with window 500.

| | R = 20 | | | | R = 40 | | | | R = 60 | | | |
|---|---|---|---|---|---|---|---|---|---|---|---|---|
| | **Mean** | **Min** | **Max** | **HE** | **Mean** | **Min** | **Max** | **HE** | **Mean** | **Min** | **Max** | **HE** |
| Eastern European markets/energy | | | | | | | | | | | | |
| DCC | 0.2167 | −0.1182 | 1.2758 | 0.0997 | 0.2161 | −0.1112 | 1.2690 | 0.0995 | 0.2157 | −0.1182 | 1.2690 | 0.0993 |
| ADCC | 0.2257 | 0.0876 | 1.2118 | 0.1032 | 0.2253 | −0.0403 | 1.2056 | 0.1031 | 0.2253 | −0.0353 | 1.2056 | 0.1029 |
| GO-GARCH | 0.1824 | −0.1766 | 4.8658 | 0.0436 | 0.1461 | −0.4646 | 1.7492 | 0.0431 | 0.1789 | −0.1871 | 4.8550 | 0.0420 |
| Eastern European markets/precious metal | | | | | | | | | | | | |
| DCC | 0.2187 | −2.4259 | 1.5624 | 0.0335 | 0.2187 | −2.3736 | 1.5624 | 0.0333 | 0.2179 | −2.3802 | 1.4437 | 0.0330 |
| ADCC | 0.2190 | −1.7676 | 1.8555 | 0.0340 | 0.2194 | −1.7249 | 1.8555 | 0.0338 | 0.2175 | −1.7347 | 1.6327 | 0.0334 |
| GO-GARCH | 0.1181 | −6.3530 | 3.0662 | 0.0445 | 0.1180 | −6.3168 | 3.0663 | 0.0444 | 0.2191 | −0.9030 | 2.0137 | 0.0446 |
| Eastern European markets/industrial metals | | | | | | | | | | | | |
| DCC | 0.3745 | −0.3321 | 2.7785 | 0.1077 | 0.3724 | −0.3321 | 2.7970 | 0.1071 | 0.3723 | −0.3321 | 2.7785 | 0.1071 |
| ADCC | 0.3830 | −0.2943 | 2.6705 | 0.1132 | 0.3813 | −0.2943 | 2.7013 | 0.1126 | 0.3812 | −0.2943 | 2.6705 | 0.1127 |
| GO-GARCH | 0.2334 | −1.1252 | 3.7086 | 0.0696 | 0.2337 | −0.2182 | 0.8940 | 0.0691 | 0.2319 | −0.2184 | 0.8931 | 0.0684 |
| Eastern European markets/livestock | | | | | | | | | | | | |
| DCC | 0.1533 | −0.4636 | 3.6635 | 0.0083 | 0.1533 | −0.4636 | 3.6833 | 0.0083 | 0.1519 | −0.1838 | 3.6531 | 0.0081 |
| ADCC | 0.1489 | −0.6230 | 2.6433 | 0.0087 | 0.1495 | −0.6230 | 2.6523 | 0.0087 | 0.1487 | −0.2376 | 2.6320 | 0.0086 |
| GO-GARCH | 0.0194 | −0.4424 | 0.5042 | 0.0081 | 0.0183 | −0.4557 | 0.5002 | 0.0077 | 0.0523 | −1.5749 | 6.0137 | 0.0074 |
| Eastern European markets/agriculture | | | | | | | | | | | | |
| DCC | 0.1861 | −0.4592 | 2.6393 | 0.0247 | 0.1864 | −0.5457 | 2.6393 | 0.0247 | 0.1856 | −0.5457 | 2.4592 | 0.0247 |
| ADCC | 0.2063 | −0.3895 | 2.2641 | 0.0277 | 0.2076 | −0.2435 | 2.2641 | 0.0277 | 0.2053 | −0.3895 | 2.1712 | 0.0275 |
| GO-GARCH | 0.0729 | −0.8797 | 0.3178 | 0.0260 | 0.0717 | −0.8804 | 0.3179 | 0.0254 | 0.0710 | −0.8814 | 0.3116 | 0.0253 |

**Table 9.** Hedge-ratio summary statistics and hedging effectiveness (HE) with window 1500.

| | R = 20 | | | | R = 40 | | | | R = 60 | | | |
|---|---|---|---|---|---|---|---|---|---|---|---|---|
| | **Mean** | **Min** | **Max** | **HE** | **Mean** | **Min** | **Max** | **HE** | **Mean** | **Min** | **Max** | **HE** |
| Eastern European markets/energy | | | | | | | | | | | | |
| DCC | 0.2517 | −0.1182 | 1.2758 | 0.1171 | 0.2512 | −0.1112 | 1.2690 | 0.1170 | 0.2510 | −0.1112 | 1.2572 | 0.1169 |
| ADCC | 0.2621 | −0.0876 | 1.2118 | 0.1213 | 0.2617 | −0.0403 | 1.2056 | 0.1212 | 0.2614 | −0.0876 | 1.1940 | 0.1210 |
| GO-GARCH | 0.1987 | −0.0607 | 4.8658 | 0.0493 | 0.1971 | −0.0612 | 4.8683 | 0.0488 | 0.1969 | −0.0607 | 4.8672 | 0.0483 |
| Eastern European markets/precious metal | | | | | | | | | | | | |
| DCC | 0.2265 | −2.4259 | 1.4802 | 0.0385 | 0.2266 | −2.3736 | 1.4470 | 0.0382 | 0.2270 | −2.4259 | 1.4802 | 0.0381 |
| ADCC | 0.2251 | −1.7676 | 1.2032 | 0.0390 | 0.2253 | −1.7249 | 1.2192 | 0.0387 | 0.2257 | −1.7676 | 1.2162 | 0.0387 |
| GO-GARCH | 0.2474 | −0.3845 | 2.0149 | 0.0497 | 0.1177 | −6.3168 | 3.0663 | 0.0496 | 0.2466 | −0.3823 | 1.9980 | 0.0499 |
| Eastern European markets/industrial metals | | | | | | | | | | | | |
| DCC | 0.4259 | −0.3321 | 2.7785 | 0.1260 | 0.4254 | −0.3321 | 2.7970 | 0.1254 | 0.4255 | −0.3375 | 2.7970 | 0.1254 |
| ADCC | 0.4351 | −0.2943 | 2.6705 | 0.1323 | 0.4347 | −0.2943 | 2.7013 | 0.1318 | 0.4351 | −0.30146 | 2.7013 | 0.1319 |
| GO-GARCH | 0.2556 | −0.0303 | 3.2698 | 0.0789 | 0.2727 | −0.1175 | 0.8940 | 0.0782 | 0.2732 | −0.1119 | 0.8872 | 0.0794 |

**Table 9.** *Cont.*

| | R = 20 | | | | R = 40 | | | | R = 60 | | | |
|---|---|---|---|---|---|---|---|---|---|---|---|---|
| | **Mean** | **Min** | **Max** | **HE** | **Mean** | **Min** | **Max** | **HE** | **Mean** | **Min** | **Max** | **HE** |
| Eastern European markets/livestock | | | | | | | | | | | | |
| DCC | 0.1751 | −0.1979 | 3.6635 | 0.0096 | 0.1750 | −0.1838 | 3.6833 | 0.0096 | 0.1734 | −0.1901 | 3.6635 | 0.0094 |
| ADCC | 0.1702 | −0.6230 | 2.6433 | 0.0101 | 0.1708 | −0.6230 | 2.6523 | 0.0101 | 0.1696 | −0.5546 | 2.6433 | 0.0098 |
| GO-GARCH | 0.0243 | −0.44242 | 0.5042 | 0.0092 | 0.0686 | −0.2454 | 6.0522 | 0.0086 | 0.0770 | −0.4324 | 6.0499 | 0.0107 |
| Eastern European markets/agriculture | | | | | | | | | | | | |
| DCC | 0.1986 | −0.2021 | 1.3834 | 0.0284 | 0.1985 | −0.1988 | 1.3791 | 0.0283 | 0.1965 | −0.1988 | 1.3791 | 0.0278 |
| ADCC | 0.2146 | −0.1874 | 1.6519 | 0.0316 | 0.2144 | −0.1730 | 1.6426 | 0.0316 | 0.2135 | −0.1874 | 1.6426 | 0.0311 |
| GO-GARCH | 0.1321 | −0.1120 | 2.8540 | 0.0281 | 0.0767 | −0.8804 | 0.3179 | 0.0277 | 0.0764 | −0.8895 | 0.3178 | 0.0291 |

**Table 10.** Hedge-ratio summary statistics and hedging effectiveness (HE) with window 2000.

| | R = 20 | | | | R = 40 | | | | R = 60 | | | |
|---|---|---|---|---|---|---|---|---|---|---|---|---|
| | **Mean** | **Min** | **Max** | **HE** | **Mean** | **Min** | **Max** | **HE** | **Mean** | **Min** | **Max** | **HE** |
| Eastern European markets/energy | | | | | | | | | | | | |
| DCC | 0.2748 | −0.1182 | 1.2758 | 0.1284 | 0.2745 | −0.1182 | 1.2758 | 0.1283 | 0.2739 | −0.1182 | 1.2690 | 0.1280 |
| ADCC | 0.2859 | −0.0876 | 1.2118 | 0.1330 | 0.2858 | −0.0876 | 1.2118 | 0.1329 | 0.2854 | −0.0321 | 1.2056 | 0.1327 |
| GO-GARCH | 0.2177 | −0.0512 | 4.8658 | 0.0538 | 0.1906 | −0.4588 | 1.7405 | 0.0530 | 0.2126 | −0.0477 | 4.8550 | 0.0516 |
| Eastern European markets/precious metal | | | | | | | | | | | | |
| DCC | 0.2452 | −2.4259 | 1.4802 | 0.0420 | 0.2458 | −2.4259 | 1.4802 | 0.0419 | 0.2448 | −2.3802 | 1.4437 | 0.0415 |
| ADCC | 0.2450 | −1.7676 | 1.2032 | 0.0426 | 0.2456 | −1.7676 | 1.2032 | 0.0425 | 0.2443 | −1.7347 | 1.2192 | 0.0419 |
| GOGARCH | 0.1204 | −6.3530 | 3.0662 | 0.0539 | 0.1077 | −6.3530 | 0.5842 | 0.0541 | 0.2659 | −0.3842 | 2.0137 | 0.0540 |
| Eastern European markets/industrial metals | | | | | | | | | | | | |
| DCC | 0.4474 | −0.3321 | 2.7785 | 0.1370 | 0.4473 | −0.3375 | 2.7785 | 0.1368 | 0.4467 | −0.3321 | 2.7785 | 0.1364 |
| ADCC | 0.4585 | −0.2943 | 2.6705 | 0.1442 | 0.4586 | −0.3014 | 2.6705 | 0.1441 | 0.4579 | −0.2943 | 2.6705 | 0.1436 |
| GO-GARCH | 0.2949 | −0.1259 | 0.9140 | 0.0858 | 0.2936 | −0.0711 | 0.9126 | 0.0862 | 0.2678 | −0.0235 | 3.2414 | 0.0842 |
| Eastern European markets/livestock | | | | | | | | | | | | |
| DCC | 0.1900 | −0.1145 | 3.6635 | 0.0105 | 0.1887 | −0.1139 | 3.6635 | 0.0103 | 0.1879 | −0.1139 | 3.6531 | 0.0103 |
| ADCC | 0.1847 | −0.6230 | 2.6433 | 0.0109 | 0.1831 | −0.1911 | 2.6433 | 0.0107 | 0.1837 | −0.1911 | 2.6320 | 0.0108 |
| GO-GARCH | 0.0803 | −0.2291 | 6.0514 | 0.0101 | 0.0835 | −0.4084 | 6.0509 | 0.0112 | 0.0249 | −0.4559 | 0.4686 | 0.0092 |
| Eastern European markets/agriculture | | | | | | | | | | | | |
| DCC | 0.2095 | −0.2021 | 1.3834 | 0.0309 | 0.2082 | −0.2021 | 1.3834 | 0.0306 | 0.2091 | −0.2021 | 1.3834 | 0.0309 |
| ADCC | 0.2239 | −0.1874 | 1.6519 | 0.0343 | 0.2231 | −0.1874 | 1.6519 | 0.0341 | 0.2222 | −0.1807 | 1.6519 | 0.0341 |
| GO-GARCH | 0.1380 | −0.1120 | 2.8540 | 0.0303 | 0.1384 | −0.1135 | 2.8518 | 0.0310 | 0.0770 | −0.8814 | 0.3116 | 0.0293 |

**Table 11.** The top hedging alternatives according to the indicator (HE) relevant to the different refits and windows.

| Window | Refit | The 1st Sector According to the Maximum HE Criterion | HE | The 2nd Sector According to the Maximum HE Criterion | HE |
|---|---|---|---|---|---|
| 500 | 20 | Industrial metals | 11.32% | Energy | 10.32% |
| | 40 | Industrial metals | 11.26% | Energy | 10.31% |
| | 60 | Industrial metals | 11.27% | Energy | 10.29% |
| 1500 | 20 | Industrial metals | 13.23% | Energy | 12.13% |
| | 40 | Industrial metals | 13.18% | Energy | 12.12% |
| | 60 | Industrial metals | 13.19% | Energy | 12.10% |
| 2000 | 20 | Industrial metals | 14.42% | Energy | 13.30% |
| | 40 | Industrial metals | 14.41% | Energy | 13.29% |
| | 60 | Industrial metals | 14.36% | Energy | 13.27% |

## 6. Robustness Analysis

This section is devoted to highlighting the obtained robustness results of hedging effectiveness under specific assumptions concerning the distribution choice of the MGARCH models, the number of model refits, and the length of the forecast horizon. These results are reported in Tables 8–10, which represent the hedging ratios and effectiveness for each pair between the Eastern European markets indices and commodities sectors indices; distributed with a skewed Student distribution; estimated with the DCC, ADCC and GO-GARCH models; for 20, 40 and 60 daily refits; and using the length of one-step forecasts horizon 500, 1500 and 2000, respectively.

The empirical findings show that the ADCC model supplies the highest hedging effectiveness in contrast to the GO-GARCH model, which is the least effective in the majority of cases. On the other hand, the hedging effectiveness ($HE$) values produced by the DCC and ADCC models are very similar in all cases. A possible explanation of these results is that both of them (the DCC and ADCC models) capture data properties in a similar way, but the ADCC model is more reliable since it captures asymmetric effects.

Concerning the level of refit for both models, they have the same level of ($HE$) across the different model refits. For example, the Eastern European markets/energy hedge pair for the ADCC model appears to result in ($HE$) values of 0.1032, 0.1031 and 0.1029 for 20, 40 and 60 daily refits, respectively (Table 8).

On the other hand, the forecast length or number of one-step forecasts of 500 starts from 29 June 2020, the forecast length of 1500 starts from 28 September 2016 and the forecast length of 2000 starts from 29 October 2014, respectively. These periods cover several extreme events and turbulence in the financial markets followed by the Russo-Ukrainian war (2022), the COVID-19 global pandemic (2019–2020), the commodities price collapse (2015–2016) and China's stock market crisis (2015). The results show that for a longer forecast length, the hedging effectiveness values increase: for example, the Eastern European markets/energy hedge pair for the ADCC model records ($HE$) values of 0.1032, 0.1213 and 0.1330 for 500, 1500 and 2000 forecast length, respectively (Table 8).

## 7. Conclusions and Implications

In this paper, a thorough investigation was carried out. It concerns the optimal hedging emerging Eastern Europe stock markets with commodities sectors during periods of major financial turbulence, including the COVID-19 pandemic period and the Russian-Ukrainian war period. We implemented the DCC, the ADCC and the GO-GARCH models to estimate one-step-ahead of the dynamic conditional correlation forecasts and then we calculated hedge ratios and hedging effectiveness. After comparing hedging effectiveness, what can be inferred from this paper is that industrial metals represent the most attractive alternative sector as long as it displays the highest hedging effectiveness values compared to other sectors, among all the alternative sectors. The second-highest hedging effectiveness values are illustrated in the energy sector. In addition, the hedge ratios registered from the DCC and ADCC, which record the highest hedging effectiveness values, are closely similar, while those from the GO-GARCH model are different. It is noteworthy that our findings are robust in terms of model daily refits (every 20, 40 and 60) and forecast lengths of 500, 1500 and 2000; we also find that a longer forecast length increases the hedging effectiveness values. Finally, the hedge ratios change significantly during the sample period, which means that hedge ratios must be updated regularly.

In fact, our results are of great interest to investors searching for efficient hedging strategies. Our strong, yet modest, recommendation is to use the ADCC model to measure hedging effectiveness thanks to the fact that ADCC records the highest value with regard to all observed cases, rather than using the GO-GARCH model because it records a lower value in many cases. A further recommendation is to hedge emerging Eastern Europe stock markets with the industrial metals sector since it is the best strategy to reduce risks.

**Author Contributions:** All authors have contributed equally to this mansucript. All authors have read and agreed to the published version of the manuscript.

**Funding:** This research received no external funding.

**Institutional Review Board Statement:** Not applicable.

**Informed Consent Statement:** Not applicable.

**Data Availability Statement:** The data presented in this study were collected from the DataStream database.

**Conflicts of Interest:** The authors declare no conflict of interest.

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
