# Peer review of "Which Commodity Sectors Effectively Hedge Emerging Eastern European Stock Markets? Evidence from MGARCH Models"

_commodities, doi:10.3390/commodities2030016_

Round 1
Reviewer 1 Report
The paper contains a nice application of selected mGARCH-type models to risk management of Eastern European stock investment. The research problem is well motivated and well grounded in the literature (a comprehensive review). The design of empirical analysis seems rather sound and the conclusions are well justified. However, the paper is difficult to follow since its quality is seriously undermined by careless editing and poor presentation (varying font sizes, messy formula and table placement, an excessive use of overly long tables and charts). Unless a serious attempt is made to improve this aspect and make the paper substantially more reader friendly, it is not suitable for publication.
The language is essentially OK, but in places there are some noticeable linguistic errors that seriously affect the outcome. Just three examples:
1. p1. l. 8: " to construct a forward step out of sample"
2. p2, l. 53: "It is obvious that the hedging strategy is completely different over time." ?? perhaps varies (or should vary) over time...
3. p. 23, l. 558-9: "the comparison of hedging effectiveness shows that Industrial Metals is the most attractive alternative sector as long as it displays the highest hedging effectiveness values"
Proofreading strongly recommended.
Author Response
|
|
Reviewer’s Feedback |
Author(s)’ Response/Action |
|
Comments and Suggestions for Authors |
[However, the paper is difficult to follow since its quality is seriously undermined by careless editing and poor presentation (varying font sizes, messy formula and table placement, an excessive use of overly long tables and Charts) ] |
-I resized the fonts to be standard and I relocated table 4. -I changed the formula to standard. -As far as the long tables are concerned, the nature of this article requires the presence of such tables in order to be comprehensive. This is the same thing for the figures.
|
|
Comments on the Quality of English Language |
[Comments on the Quality of English Language The language is essentially OK, but in places there are some noticeable linguistic errors that seriously affect the outcome. Just three examples: 1. p1. l. 8: " to construct a forward step out of sample" 2. p2, l. 53: "It is obvious that the hedging strategy is completely different over time." ?? perhaps varies (or should vary) over time... 3. p. 23, l. 558-9: "the comparison of hedging effectiveness shows that Industrial Metals is the most attractive alternative sector as long as it displays the highest hedging effectiveness values"] |
As an attempt to improve the quality of the English language, I corrected the suggested sentences ("p1. l. 8" , "p2, l. 53" ,and "p. 23, l. 558-9")
|

Reviewer 2 Report
The paper demonstrates the use of sophisticated statistical methods to analyze and predict hedging effectiveness.
The result of the article is the proposal of the ADCC model and strategy based on the metal sector. The statistical study was accompanied by an analysis of the appropriate model refitting period.
I have only two more specific comments:
1) In the article, I did not understand how to set up the Q0 matrix and how to estimate the parameters theta1 and theta2 in formula (8).
2) The authors pay attention to the robustness of the models. But for me, the question remains, what is the resistance of the methods to changes in correlation matrices over time?
Minor typos:
L10 December1994 -> December 1994
L14 the hedging effectiveness are/is
L24 amongst/between
Author Response
|
Comments and Suggestions for Authors (1) |
[1) In the article, I did not understand how to set up the Q0 matrix and how to estimate the parameters theta1 and theta2 in formula (8).] |
-The positive parameters theta1and theta2 are associated with the exponential smoothing process. The theta1and theta2 are utilized to produce the dynamic conditional correlations. -The symmetric positive matrix ( Q) in formula (8) set up by: -( theta1and theta2) then so (1- (theta1+ theta2)> 0) - (Q bar matrix) illustrates the unconditional correlation matrix of the standardized residuals -Q_(t-1) are symmetric positive matrix at time t-1
|
|
Comments and Suggestions for Authors (2) |
[ The authors pay attention to the robustness of the models. But for me, the question remains, what is the resistance of the methods to changes in correlation matrices over time?] |
-I changed the correlation matrices over time to produce a new hedge ratio, and then build a new hedging strategy taking into account the recent information. I thought that producing a dynamic parameter is better than static parameters which do not take into account the recent information. |

Reviewer 3 Report
The theme of financial markets and commodities are topics that are on the agenda, always current. For mature, efficient, and functioning financial markets are a foundation for structured and functioning economies. In this sense, I consider that the theme is relevant and current.
The work is well structured. The title fits and is representative of the work, the abstract, the introduction, the literature review, the formulation of hypotheses, and the discussion of results, are written in a thoughtful and balanced way.
The methodology is presented clearly and objectively, I consider it to be the right one to achieve the objectives of the work, the discussion and presentation of results are also carried out in a clear, comprehensive, and objective way.
The conclusion is presented in an objective and parsimonious way, without being long and tedious, however, the author may refer to limitations or difficulties encountered throughout the work.
Regarding the bibliography, I suggest some changes, from the bibliography referenced throughout the work, only approximately 26% is from the last 5 years. As a proposal for improving the work, I propose that the author consider the possibility of increasing this ratio, to a value of no less than 30%. The author in the references does not follow a homogeneous rule throughout the references, for example:
- Presents the DOI in a minority of works, the author must choose to follow a defined rule, present it to all or not present it;
- The year of reference of the elaboration of the work, sometimes it is presented between parentheses (2020) others, 2020, example: the first two, but it is repeated throughout the references;
- There are works with an indication of the year at the end of the reference.
In my opinion, the author should present the references homogeneously.
Author Response
|
|
[Regarding the bibliography, I suggest some changes, from the bibliography referenced throughout the work, only approximately 26% is from the last 5 years. As a proposal for improving the work, I propose that the author consider the possibility of increasing this ratio, to a value of no less than 30%.] |
-I have added some recent references to increase the percentage of articles published in the last five years in my paper to achieve 30% of all references are published in the last five years. View (2. Literature review) |
|
|
[The author in the references does not follow a homogeneous rule throughout the references, for example: Presents the DOI in a minority of works, the author must choose to follow a defined rule, present it to all or not present it; - The year of reference of the elaboration of the work, sometimes it is presented between parentheses (2020) others, 2020, example: the first two, but it is repeated throughout the references; - There are works with an indication of the year at the end of the reference. In my opinion, the author should present the references homogeneously.] |
I corrected all references to be homogeneous, which are written as in the below example: "Abdallah, A., Ghorbel, A. (2020), “Can alternative hedging assets add value to clean energy portfolio? Evidence from MGARCH models”, International Journal of Scientific Research & Engineering Technology (IJSET), Vol.14 pp.48-67."
|

Round 2
Reviewer 1 Report
I stick to my previous recommendation. The paper has not been substantially redrafted (and proofread) as requested and its quality is still well below acceptable level.
See above.
Author Response
|
Reviewer’s Feedback |
Author(s)’ Response/Action |
|
| Comments and Suggestions for Authors |
I stick to my previous recommendation. The paper has not been substantially redrafted (and proofread) as requested and its quality is still well below acceptable level. |
-I relocated all the tables and figures. -I changed the formula to be well presented -As far as the long tables are concerned, I rewrited the tables 3 and 4 making them clearer and less voluminous. -I modified the methods to be more descriptive. -I modified the description of the results to be clearer. |
| Comments on the Quality of English Language |
1) See above |
-I tried to improve the quality of English hoping that it would be better. |

Round 3
Reviewer 1 Report
Dear Author(s), the paper is now in the form more or less suitable for a review. It is however still far from perfect mainly in terms of form (occasional linguistic errors, punctuation missing, typos, difficult to comprehend tables and charts, etc.) and also substance in places. Had you submitted it in a better shape, it would have benefited much more from the review process...
Still a room for improvement.